# PeerJ

# Comparison of automated nucleic acid extraction methods for the detection of cytomegalovirus DNA in fluids and tissues

Jesse J. Waggoner[1] and Benjamin A. Pinsky[1,2]

[1] Department of Medicine, Division of Infectious Diseases and Geographic Medicine, Stanford University School of Medicine, Stanford, CA, USA
[2] Department of Pathology, Stanford University School of Medicine, Stanford, CA, USA

## ABSTRACT

Testing for cytomegalovirus (CMV) DNA is increasingly being used for specimen types other than plasma or whole blood. However, few studies have investigated the performance of different nucleic acid extraction protocols in such specimens. In this study, CMV extraction using the Cell-free 1000 and Pathogen Complex 400 protocols on the QIAsymphony Sample Processing (SP) system were compared using bronchoalveolar lavage fluid (BAL), tissue samples, and urine. The QIAsymphonyAssay Set-up (AS) system was used to assemble reactions using *artus* CMV PCR reagents and amplification was carried out on the Rotor-Gene Q. Samples from 93 patients previously tested for CMV DNA and negative samples spiked with CMV AD-169 were used to evaluate assay performance. The Pathogen Complex 400 protocol yielded the following results: BAL, sensitivity 100% (33/33), specificity 87% (20/23); tissue, sensitivity 100% (25/25), specificity 100% (20/20); urine, sensitivity 100% (21/21), specificity 100% (20/20). Cell-free 1000 extraction gave comparable results for BAL and tissue, however, for urine, the sensitivity was 86% (18/21) and specimen quantitation was inaccurate. Comparative studies of different extraction protocols and DNA detection methods in body fluids and tissues are needed, as assays optimized for blood or plasma will not necessarily perform well on other specimen types.

Corresponding author
Benjamin A. Pinsky,
bpinsky@stanford.edu

## INTRODUCTION

Cytomegalovirus (CMV) is a common human viral pathogen with anywhere from 45%–100% of adults demonstrating serologic evidence of CMV exposure, depending on the population (*Cannon, Schmid & Hyde, 2010*). CMV infection remains an important cause of morbidity and mortality following solid organ and hematopoietic stem cell transplantation (*Kotton et al., 2010*; *Ljungman et al., 2008*). CMV disease has also been shown to predict treatment failure in ulcerative colitis (UC), and it can cause sensorineural hearing loss and developmental abnormalities in congenitally infected neonates, even when asymptomatic at birth (*Lanari et al., 2006*; *Roblin et al., 2011*).

Quantitative PCR testing for CMV from patient plasma or whole blood has become the standard of care for detection and monitoring in transplant recipients (*Caliendo et al., 2007*; *Kotton et al., 2010*). The definitive diagnosis of CMV end-organ disease in immunocompromised patients often requires demonstration of characteristic viral inclusions or detection of viral proteins by immunohistochemical staining in tissue samples. However, PCR testing for CMV DNA in body fluids and tissues is increasingly being used to aid in the diagnostic work-up for entities such as CMV pneumonitis in lung transplant recipients or colonic involvement in UC (*Ljungman, Griffiths & Paya, 2002*; *Riise et al., 2000*; *Roblin et al., 2011*). In the case of congenital CMV infection, diagnosis is usually made by detection of the virus in urine or saliva, although prenatal testing using amniotic fluid samples in quantitative PCR or RT-PCR assays has been described (*Goegebuer et al., 2009*; *Lanari et al., 2006*; *Revello et al., 2003*).

A number of comparative analyses of methods for CMV DNA detection and quantitation in plasma have been published, but similar evaluations of the detection of CMV DNA in non-plasma specimens are limited (*Bravo et al., 2011*; *Caliendo et al., 2007*; *Fahle & Fischer, 2000*; *Forman, Wilson & Valsamakis, 2011*). Furthermore, it is unclear if the same nucleic acid extraction protocol used to isolate CMV DNA from plasma will be the optimum for non-plasma samples (*Fahle & Fischer, 2000*; *Tang et al., 2005*; *Verheyen et al., 2012*). In this study, we describe the performance characteristics of the *artus* CMV PCR for the qualitative detection of CMV DNA in patient bronchoalveolar lavage (BAL) fluid, tissue samples, and urine using two automated extraction protocols, the Cell-free 1000 and Pathogen Complex 400, on the QIAsymphony Sample Processing (SP)/Assay Set-up (AS) system and Rotor-Gene Q (RGQ). The Cell-free 1000 protocol is a plasma extraction protocol, whereas the Pathogen Complex 400 was designed for use with serum, plasma, CSF, respiratory specimens (including BAL fluid), and urine. These results are compared with those obtained using a previous laboratory standard, the COBAS Amplicor CMV Monitor (CACM) assay following extraction with the MagNA Pure LC DNA isolation kit.

## MATERIALS AND METHODS

### Specimens

The Stanford University Institutional Review Board waived approval for this work as it was performed in the clinical virology laboratory for the purposes of quality assurance and test validation.

Ninety-three archived patient specimens, previously tested using the CACM assay (Roche, Indianapolis, IN), were evaluated. Samples were stored as aliquots at −80 °C following initial testing. These specimens included 36 BAL (13 detected, 23 not detected), 25 tissues [5 detected (3 small bowel, 2 colon), 20 not detected (5 skin, 4 small bowel, 4 colon, 3 liver, 2 lung, 1 lymph node, 1 anterior mediastinal mass], and 21 urine samples (1 detected, 20 not detected). BAL specimens were collected in sterile containers, and if mucoid, the samples were vortexed with sterile glass beads to reduce specimen viscosity. Tissue specimens were collected in M4RT (Remel, Lenexa, KS) viral transport media (VTM) or sent in a sterile container, and upon receipt, transferred to Bartels Tissue

Culture Refeeding Media (Trinity Biotech USA, Inc., Jamestown, NY) supplemented with 2% fetal bovine serum, 15 μg/ml gentamicin, 100 μg/ml vancomycin, and 3.5 μg/ml amphotericin B. Tissues were minced using a sterile scalpel and then ground with a tissue grinder.

Quantitated cell culture lysate, obtained from fibroblast cells infected with the AD-169 strain of CMV (ATCC VR-538), was used to spike negative samples for extraction using the different protocols. Cell culture lysate was quantitated using the *artus* CMV PCR. For BAL and tissue specimens, 10-fold dilutions were prepared wherein samples were spiked to a final concentration of 6.6 to 2.6 $\log_{10}$ copies/ml. Four samples of each specimen type were tested at each concentration. The concentration for tissue samples was calculated using the volume of media in each sample after the sample was ground. For the urine samples, four samples were tested at 10-fold dilutions extending from 6.0 $\log_{10}$ to 2.0 $\log_{10}$ copies/ml. Patient specimens that had previously tested positive are referred to as clinical samples to differentiate them from spiked samples.

## Nucleic acid extraction and assay set-up

For the reference method, 100 μl of each specimen was extracted on the MagNA Pure LC (Roche, Indianapolis, IN) using the MagNA Pure LC DNA isolation kit with the DNA I Blood Cell High Performance protocol. The CACM assay was set-up manually using 50 μl out of the 100 μl elution.

For the test methods, 400 μl of specimen was extracted on the QIAsymphony SP (Qiagen, Germantown, MD) using the Qiagen Virus/Bacteria Midi kit and Pathogen Complex 400 protocol. Specimens were also extracted using the same kit and the Cell-free 1000 protocol, which extracts from 1000 μl of sample. For eight negative tissue specimens, there was insufficient volume for extraction using the Cell-free 1000 protocol. The *artus* CMV PCR assay (Qiagen, Germantown, MD) was set-up with a custom protocol on the QIAsymphony AS that utilizes 10 μl of the 95 μl elution, 12.5 μl CMV RG Master, and 2.5 μl of CMV Mg-Sol, for a final reaction volume of 25 μl.

## CMV amplification and detection

The CACM reactions utilized non-saturating, end-point PCR targeting the CMV polymerase UL54 gene followed by detection via an automated enzyme-linked oligosorbent assay (ELOSA). The *artus* CMV RG PCR reactions used a real-time, hydrolysis-probe-based PCR targeting the CMV Major Immediate Early gene and were run on the RGQ. The reactions underwent 10 min at 95 °C, then 10 cycles of touchdown PCR with the annealing step starting at 65 °C for 30 s and decreasing by 1 °C each cycle. Denaturation was at 95 °C for 15 s and extension at 72 °C for 20 s. Touchdown PCR was followed by 35 cycles of 95 °C for 15 s, 56 °C for 30 s, and 72 °C for 20 s. Data was collected on the green and yellow channels. A four-point standard curve was included in each run. Amplification of an internal control added prior to extraction ensured adequate extraction efficiency and the absence of inhibitors. The analytical performance of the *artus* CMV PCR has been described previously (*Waggoner et al., 2012*).

## Statistics

Basic statistical analysis was performed using Excel software (Microsoft, Redmond, WA). Paired t-tests were performed using GraphPad software (GraphPad, La Jolla, CA). Shapiro–Wilk tests were performed using SPSS software (IBM, Armonk, NY) to confirm the normality of the distribution of the differences in the quantified CMV viral load.

# RESULTS AND DISCUSSION

## Bronchoalveolar lavage fluid

Thirty-six BAL specimens originally extracted with the MagNA Pure LC and tested on the CACM were subsequently analyzed with the *artus* CMV RG PCR reagents on the RGQ after automated extraction and assay set-up using the QIAsymphony SP/AS. An additional 20 negative BAL specimens were spiked with AD-169 and tested using the QIAsymphony SP/AS/RGQ. The *artus* reagents demonstrated 100% (33/33) sensitivity when samples were extracted with either the Cell-free 1000 or Pathogen Complex 400 protocols. This included 13/13 clinical samples and 20/20 spiked samples. Though this protocol was developed for qualitative testing on non-plasma specimen types, quantified results were generated and expressed as copies/ml. The *artus* assay was linear over the range of concentrations tested following the extraction of spiked specimens with either protocol (Figs. 1A and 1B), and no significant difference in the quantitative results was observed following extraction with either protocol (mean difference 0.05, 95% CI [−0.03 to 0.13], $p = 0.19$).

The specificity for the *artus* reagents with the Cell-free 1000 and Pathogen Complex 400 protocols were 83% (19/23) and 87% (20/23), respectively. One sample was detected only using the Cell-free 1000 protocol at a very low level (~23 copies/ml). This specimen was negative for CMV by viral culture, though the patient had been previously exposed to CMV (CMV IgG positive). Additionally, there were three discrepant samples obtained from separate patients that were detected by both the Pathogen Complex 400 and Cell-free 1000 protocols, but not CACM. One of these patients received a CMV positive hematopoietic stem cell transplant and had a history of CMV hepatitis and retinitis. CMV was quantitated in this BAL specimen at 1062 copies/ml using Pathogen Complex 400 and 472 copies/ml using Cell-free 1000. There was limited clinical and laboratory data available for the remaining two patients. Only one had CMV serologic testing performed at our institution (CMV IgG positive) and for both patients concurrent BAL viral cultures were negative for CMV. Quantitation of CMV DNA in these specimens revealed 112 and 29 copies/ml using Pathogen Complex 400 and 203 and 36 copies/ml using Cell-free 1000.

## Tissue

Twenty-five previously tested tissue specimens and 20 negative tissue specimens spiked with AD-169 were analyzed on the QIAsymphony SP/AS/RGQ system. The *artus* reagents demonstrated 100% sensitivity for tissue specimens extracted using either the Cell-free 1000 or Pathogen Complex 400 protocols (25/25, including 5/5 clinical samples and 20/20 spiked samples). The specificity was 100% following extraction with either protocol,

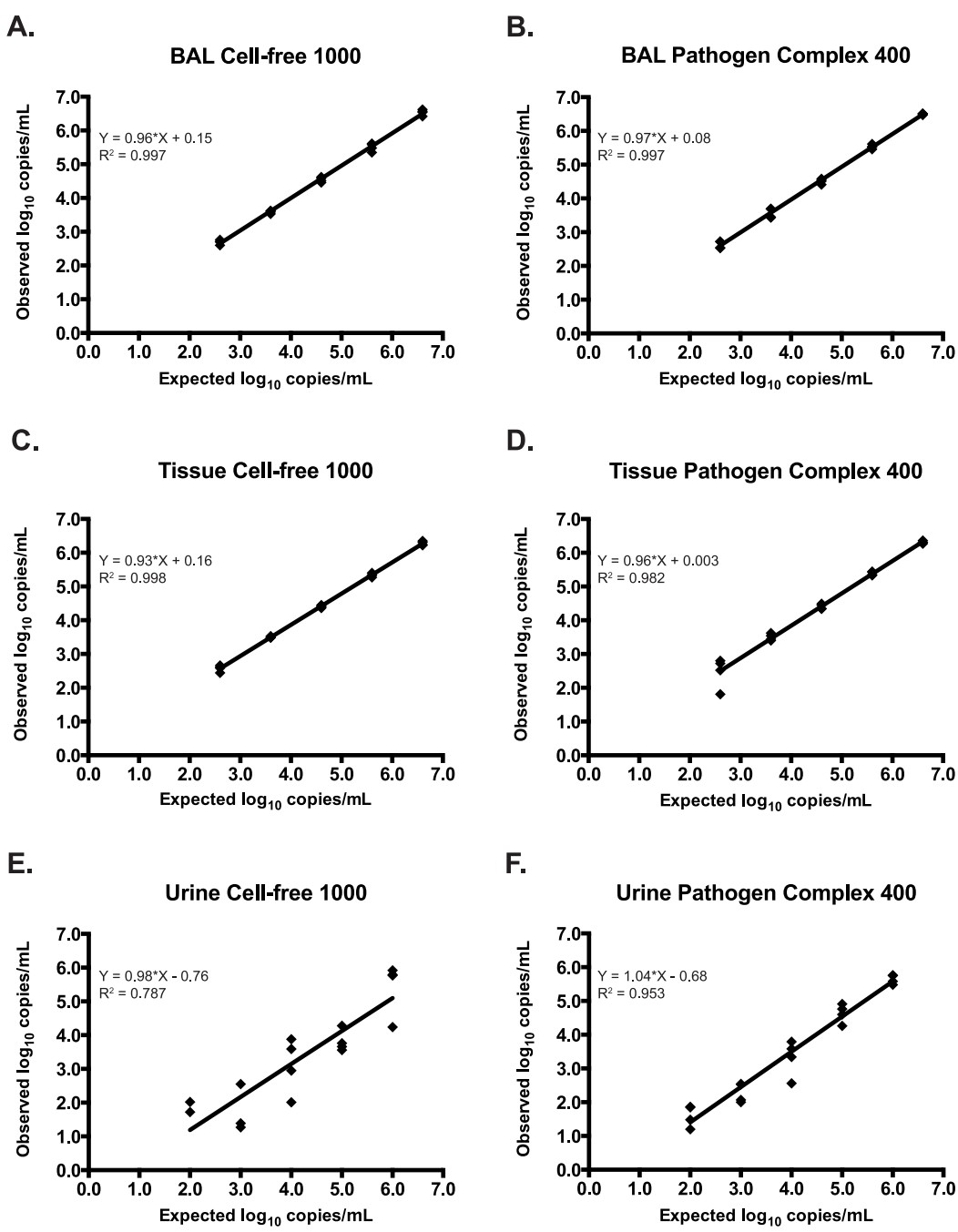

**Figure 1 Assay linearity by specimen type using different extraction protocols.** Linearity of the *artus* CMV RG PCR assay on DNA extracted from negative clinical samples spiked with serial dilutions of AD-169 using the Cell-free 1000 and Pathogen Complex 400 protocols on the QIAsymphony SP: BAL samples (A and B), tissue samples (C and D), and urine samples (E and F).

though only a subset of samples could be tested by both protocols (Cell-free 1000, 12/12; Pathogen Complex 400, 20/20). Quantitation of spiked tissue specimens using *artus* reagents showed good agreement with the expected results following extraction with both protocols, and the assay was linear over the range of concentrations tested (Figs. 1C and 1D). No difference in the quantitative results obtained using either extraction protocol was seen for the tissue samples (mean difference 0.003, 95% CI [$-0.09$ to 0.1], $p = 0.94$).

## Urine

Twenty-one previously tested urine specimens and 20 negative urine specimens spiked with AD-169 were analyzed on the QIAsymphony SP/AS/RGQ system. Pathogen Complex 400 extraction demonstrated 100% sensitivity (21/21, including one clinical sample and 20/20 spiked samples) and 100% specificity (20/20), whereas Cell-free 1000 extraction showed 86% sensitivity (18/21, including one clinical sample and 17/20 spiked samples) and 100% specificity (20/20). The three spiked urine samples that tested negative following extraction with the Cell-free 1000 protocol had concentrations of 2.0 $\log_{10}$ (two samples) and 3.0 $\log_{10}$ copies/ml (one sample). Quantitation of the spiked samples extracted with the Cell-free 1000 protocol revealed poor agreement with the expected values (Fig. 1E). In contrast, quantitation following Pathogen Complex 400 extraction was linear (Fig. 1F). The concentration of CMV in the urine samples was, on average, 0.55 $\log_{10}$ copies/ml higher using the Pathogen Complex 400 extraction protocol compared to the Cell-free 1000 (95% CI [0.14 to 0.95], $p = 0.01$). $C_T$ values for the internal control occurred later following extraction using the Cell-free 1000 protocol (mean, 27.76; standard deviation, 2.27) compared to the Pathogen Complex 400 (mean, 25.08; standard deviation, 0.19; $p < 0.0001$).

CMV detection in body fluid and tissue samples is important for the diagnosis of CMV disease, yet limited comparative data exist in the literature for viral nucleic acid extraction and PCR protocols in such samples (*Chemaly et al., 2004*; *Fahle & Fischer, 2000*; *Kotton et al., 2010*). In this study, we used BAL fluid, tissue samples, and urine to evaluate two DNA extraction protocols utilizing the QIAsymphony SP/AS followed by real-time PCR with *artus* CMV PCR reagents on the RGQ. Both protocols showed good agreement with CACM, the reference method, for BAL and tissue specimens.

Quantitative testing has been reported in the literature for all of these specimen types (*Chemaly et al., 2004*; *Ganzenmueller et al., 2009*; *Kearns et al., 2002*; *Riise et al., 2000*; *Westall et al., 2004*). However, clinically meaningful thresholds for the quantitative detection CMV DNA in such specimens have not yet been defined. Even with the use of a common calibrator such as the 1st World Health Organization International Standard for Human Cytomegalovirus for Nucleic Acid Amplification Techniques, significant concerns remain regarding sample collection and processing, which cannot be standardized in a fashion similar to plasma or whole blood collection. Therefore, the testing protocols described in this study were validated in our laboratory for the qualitative detection of CMV from non-plasma tissue specimens. Using spiked clinical specimens, quantitative CMV PCR using *artus* CMV reagents allowed for comparisons of the efficiency of DNA
extraction with the Cell-free 1000 and Pathogen Complex 400 protocols. The ranges of concentrations were selected to reflect those previously reported in the literature, though reports are scant, particularly for tissue samples (*Chemaly et al., 2004*; *de Vries et al., 2012*; *Ganzenmueller et al., 2009*; *Kearns et al., 2002*; *Riise et al., 2000*; *Roblin et al., 2011*; *Westall et al., 2004*). Following extraction with either protocol, *artus* CMV PCR was linear over the range of concentrations tested for BAL fluid and tissue samples, and the results showed good agreement with expected values.

In spiked urine samples, the Pathogen Complex 400 protocol proved more sensitive than the Cell-free 1000, and the results of the *artus* CMV PCR following extraction with the Pathogen Complex 400 were also more accurate when compared to the expected quantitative values. The detection of CMV in urine samples is primarily used for the diagnosis of congenital infection among newborns (*Lanari et al., 2006*). Despite the high viral loads often present in congenital CMV, previous reports using traditional viral culture methods have missed the diagnosis in symptomatic patients (*Nelson et al., 1995*). The severity of possible sequelae and the lack of a range in CMV viral load that defines increased risk of disease support the use of the most sensitive test to detect CMV in newborns. In addition, an extraction protocol such as Pathogen Complex 400 that allows for accurate CMV quantitation in urine may be useful in defining quantitative thresholds for risk stratification.

Based on our results, the Cell-free 1000 extraction protocol is not appropriate for urine samples as it resulted in false negatives on 3/20 spiked samples. These may have resulted from the failure of this protocol to remove PCR inhibitors from urine samples. This conclusion is supported by the finding that internal control $C_T$ values in the *artus* assay were significantly higher and showed greater variability following extraction with the Cell-free 1000 protocol compared to the Pathogen Complex 400. These two protocols differ predominantly by the inclusion of a wash step with buffer ATL (containing SDS) in the Pathogen Complex 400 protocol, and this may have accounted for the improved results in urine samples. The Cell-free 1000 protocol could still be used as the sole method for CMV DNA extraction in laboratories that do not test for CMV DNA from urine.

Three BAL samples were positive using the *artus* CMV PCR with either extraction protocol but negative by the CACM, and a single sample was positive only after extraction using the Cell-free 1000 protocol. On further evaluation, these samples appeared to have low positive results below the limit reported by the CACM. CMV PCR and hybrid capture assays have been evaluated in BAL samples from lung transplant recipients, though the clinical significance of low-positive results, such as those we detected in this study, remains unclear (*Chemaly et al., 2004*; *Riise et al., 2000*). The quantitative results from these studies have not consistently predicted CMV pneumonitis, however, no patient with a negative test for CMV had CMV pneumonitis on biopsy (negative predictive value of 100%) (*Chemaly et al., 2004*; *Riise et al., 2000*).

## CONCLUSIONS

Body fluids and tissues are important specimen types for the evaluation of CMV-related diseases. Reliable detection of CMV DNA in these specimen types depends not only on the sensitivity of the PCR assay used but also on the extraction protocol employed in the clinical laboratory. Further comparative studies of different extraction protocols and DNA detection methods in body fluids and tissues are needed, as protocols optimized for plasma or whole blood cannot always be applied to other specimen types.

## ACKNOWLEDGEMENTS

We thank the staff of the Stanford Clinical Virology Laboratory for their continued exceptional work and dedication, particularly Paolo Libiran who made important contributions to this project. Qiagen provided the *artus* CMV RG PCR reagents used in this study.

### Funding

This study was funded by the Stanford University Department of Pathology. The funders had no role in study design, data collection and analysis, decision to publish, or preparation of the manuscript.

### Grant Disclosures

The following grant information was disclosed by the authors:
Stanford University Department of Pathology.

### Competing Interests

The authors declare no competing interests in the publication of this work.

### Author Contributions

- Jesse J. Waggoner conceived and designed the experiments, performed the experiments, analyzed the data, contributed reagents/materials/analysis tools, wrote the paper, prepared figures and/or tables, reviewed drafts of the paper.
- Benjamin A. Pinsky conceived and designed the experiments, analyzed the data, contributed reagents/materials/analysis tools, wrote the paper, prepared figures and/or tables, reviewed drafts of the paper.

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
