# Peer review of "Comparison of automated nucleic acid extraction methods for the detection of cytomegalovirus DNA in fluids and tissues"

_PeerJ, doi:10.7717/peerj.334_

## Round 0.1 · original submission · Minor Revisions

Both reviewer's have suggested a number of minor clarifications which I agree need to be undertaken. In addition, please expand on the methodology undertaken in regard to the statistical testing. Was the data tested normally distributed as required for a t-test? How was normality calculated?

·

Basic reporting

Waggoner & Pinsky present here a comparison of two automated nucleic acid extraction methods for the detection of Cytomegalovirus DNA in body fluids and tissues. This topic is interesting because, as the authors explain, the methods of extraction and detection/quantitation of CMV have been developed mainly for plasma or whole blood. Most laboratories (including ours!) then apply these methods to other types of samples, which may contain CMV, after an adapted pre-treatment phase. I found the paper very clear and well written, the results are convincing enough of the analytical validity of the Pathogen Complex 400 extraction protocol and its use for CMV detection in BAL, tissue samples and urines.

I have, however, some comments and some questions:
- Introduction, line 22: It could be good to mention that whole blood is also increasingly used for CMV viral load monitoring in transplant recipients and is the gold standard in many countries.
- Introduction, line 40: For which types of samples the Pathogen complex 400 protocol was originally designed?
- Introduction, line 42: The authors should precise here that the reference method is made of the combination of both the MagNA Pure LC extraction and the CACM assay.
- Materials and Methods, line 46: Is that a prospective or a retrospective study ? Was the reference method (MagNA pure extraction + CACM) performed at the same time as the QIAsymphony + Artus analysis? This point is not clear and should be clarified. Similarly, if the QIAsymphony + Artus analysis was conducted in a retrospective way, the mode of storage of samples must be specified.
- Materials and Methods, line 51: the type of “Tissue” must be specified even if this is likely only colonic biopsies.
- Materials and Methods, line 56: Please, give the definition of the AD-169.
- Results and Discussion, lines 107-108: Sequencing of discordant samples between the reference method and QIAsymphony + Artus analysis could help to decide between true- or false-positive results.
- Results and Discussion, lines 166-174: The other advantages of using PCR for neonatal urinary diagnosis of congenital CMV infection are also the use of a smaller volume of sample than for culture and, due to its better sensitivity, the requirement of only one sample instead of three samples three consecutive days (those who work with newborns know the difficulty in collecting the urine in these patients).
- Results and Discussion, lines 174-175: I suggest to discuss a little bit more the 3 false-negative results by the Cell-free 1000 protocol. Poorer elimination of PCR inhibitors? Poorer DNA extraction than Pathogen complex 400 protocol?

Experimental design

No other comments (see above).

Validity of the findings

No other comments (see above).

Additional comments

No other comments (see above).

·

Basic reporting

No comments

Experimental design

No comments

Validity of the findings

No comments

Additional comments

Waggoner and Pinsky evaluated the artus CMV PCR assay coupled to two different automated extraction systems for the quantitation of CMV DNA in BAL, tissue specimens and urines. The data presented appear to support the authors’s conclusions. I have several comments:

1. Introduction, line 25. Definitive diagnosis require in most clinical situations the demonstration of viral CPE by IHC. Please add it.

2. Introduction, line 27. ….for entities such as CMV pneumonitis…in what patients?

3. Data on the procedence of specimens (i.e. clinical condition of patients, number of samples /patients/type of tissues….) used in this study should be provided.

4. Line 56: “Negative samples were spiked with AD169….” Was this an infected-cell culture lysate, supernatant, purified virions…?

5. Line 58: “ 10-fold dilutions were prepared to a final concentration of 6.6. to 2.6 log10/mL..” What PCR was used for CMV AD169 DNA quantitation?

6. The WHO standard should have been used for comparisons. Please discuss this issue in the corresponding section.

7. From a formal stand point it is certainly inappropriate to take as a reference method an assay which is clearly less sensitive than the test assay. Please discuss this issue.

8. Data on the characteristics of the PCR assays used in this study (LOD, LOQ, dynamic linear range….) should be included in the revised version of the manuscript.

9. Enumerate the major differences between the two extraction procedures employed

---

## Round 0.2 · accepted · Accept

Thank for your timely response to the reviewers comments and we look forward to seeing your article published in PeerJ.